

# Enhancing traditional Chinese medical named entity recognition with Dyn-Att Net: a dynamic attention approach

Jingming Hou, Saidah Saad and Nazlia Omar

Faculty of Information Science and Technology, Universiti Kebangsaan Malaysia, Bangi, Selangor, Malaysia

## ABSTRACT

Our study focuses on Traditional Chinese Medical (TCM) named entity recognition (NER), which involves identifying and extracting specific entity names from TCM record. This task has significant implications for doctors and researchers, as it enables the automated identification of relevant TCM terms, ultimately enhancing research efficiency and accuracy. However, the current Bidirectional Encoder Representations from Transformers-Long Short Term Memory-Conditional Random Fields (BERT-LSTM-CRF) model for TCM NER is constrained by a traditional structure, limiting its capacity to fully harness the advantages provided by Bidirectional Encoder Representations from Transformers (BERT) and long short term memory (LSTM) models. Through comparative experiments, we also observed that the straightforward superimposition of models actually leads to a decrease in recognition results. To optimize the structure of the traditional BERT-BiLSTM-CRF model and obtain more effective text representations, we propose the Dyn-Att Net model, which introduces dynamic attention and a parallel structure. By integrating BERT and LSTM models with the dynamic attention mechanism, our model effectively captures semantic, contextual, and sequential relations within text sequences, resulting in high accuracy. To validate the effectiveness of our model, we compared it with nine other models in TCM dataset namely the publicly available PaddlePaddle dataset. Our Dyn-Att Net model, based on BERT, outperforms the other models, achieving an F1 score of 81.91%, accuracy of 92.06%, precision of 80.26%, and recall of 83.76%. Furthermore, its robust generalization capability is substantiated through validation on the APTNER, MSRA, and EduNER datasets. Overall, the Dyn-Att Net model not only enhances NER accuracy within the realm of traditional Chinese medicine, but also showcases considerable potential for cross-domain generalization. Moreover, the Dyn-Att Net model's parallel architecture facilitates efficient computation, contributing to time-saving efforts in NER tasks.

## INTRODUCTION

Traditional Chinese medicine (TCM) serves as a time honoured cornerstone of China's healthcare system, its roots reaching back over thousands of years, enriched by extensive clinical wisdom. Central to TCM's theoretical framework are the profound concepts of

Corresponding author
Saidah Saad, saidah@ukm.edu.my

yin-yang equilibrium and the harmonious interplay of the five fundamental elements (*Long et al., 2019*). TCM perceives the human body not in isolation, but rather as an integral part of the broader natural ecosystem, emphasizing the intricate connection between human health and the external environment (*Chu et al., 2020*).

During the unprecedented global upheaval brought about by the COVID-19 pandemic, TCM emerged as a beacon of hope and healing. Its contributions in alleviating symptoms and complementing conventional medical approaches garnered significant attention and piqued the curiosity of scholars and practitioners worldwide (*Ni et al., 2020*). This newfound recognition has elevated TCM's status beyond national boundaries, making it a subject of international research and exploration.

The textual treasure trove of TCM, brimming with centuries of accumulated knowledge, holds immense value in both commercial and social spheres. Yet, the sheer volume of information within this repository poses a formidable challenge—how to efficiently sift through this wealth of text and leverage the power of artificial intelligence for information extraction and automated processing (*Ren, 2020*). This undertaking represents not only a critical component of preserving and fostering Chinese cultural heritage, but also a dynamic and ever evolving field at the forefront of global medical research. The fusion of ancient wisdom and modern technology in the context of TCM opens new avenues not only for healthcare, but also for understanding the intricate relationship between traditional practices and contemporary science. Thus, TCM embodies the essence of a living tradition, continuously adapting and shaping the healthcare landscape in profound ways.

Named entity recognition (NER) can automatically identify entities with specific meanings from text, classify and annotate them, and provide reliable information support and semantic understanding. Simultaneously, TCM NER carries substantial significance. It empowers us to automatically detect and annotate critical data within texts pertaining to traditional Chinese medicine. NER plays a pivotal role in enhancing the management of knowledge in the medical field, rendering medical related information more easily retrievable and applicable (*Yousef et al., 2020*). Furthermore, NER serves as the cornerstone for constructing specialized knowledge graph in TCM (*Osman, Noah & Saad, 2022*; *Abu-Salih et al., 2023*), thereby accelerating research in Chinese herbal medicine, the identification of potential drug candidates, and the dissemination of invaluable healthcare insights rooted in the principles of TCM.

However, in comparison to general domain NER, NER in the field of TCM faces a unique set of challenges. Firstly, TCM NER encompasses a plethora of specialized terms and named entities, such as herbal medicine, TCM treatments, and prescriptions. When these entities are labeled using the Begin, Inside, Outside (BIO) tagging scheme, it results in a more intricate and diverse set of labels. Table 1 illustrates the BIO labels employed in our study, totaling 11 distinct entity types.

Secondly, the TCM domain is replete with specialized terminologies, contributing to increased structural complexity within the text (*Liu et al., 2023a*). Consequently, the

**Table 1  TCM entity labeling form.**

| Entity type | Marking form |
| --- | --- |
| TCM diagnosis | B-TCM diagnosis, I-TCM diagnosis |
| Western medicine diagnosis | B- Western medicine diagnosis, I- Western medicine diagnosis |
| TCM treatment | B- TCM treatment, I- TCM treatment |
| Chinese herbs | B- Chinese herbs, I- Chinese herbs |
| TCM syndrome | B- TCM syndrome, I- TCM syndrome |
| Prescription | B- Prescription, I- Prescription |
| TCM treatment principles | B- TCM treatment principles, I- TCM treatment principles |
| Clinical manifestations | B- Clinical manifestations, I- Clinical manifestations |
| Western medicine treatment | B- Western medicine treatment, I- Western medicine treatment |
| Other treatments | B- Other treatments, I- Other treatments |
| Non-Entity | 0 |

**Tag:** Prescription  TCM syndrome  Western medicine diagnosis

**Text:** 柴胡达原饮治疗脾胃湿热型功能性消化不良随机对照试验

**Figure 1  An example of TCM corpus.**

process of NER becomes notably more demanding. Figure 1 presents an example from the dataset used in this research.

Lastly, prevailing NER models in the TCM domain predominantly rely on BERT-LSTM-CRF architecture due to the unique advantages offered by each component. BERT, as a pre-trained language model, excels at capturing contextual information and semantic features from text data, while long short term memory (LSTM) is adept at capturing sequential dependencies within text sequences. The conditional random field (CRF) layer, on the other hand, enables the modeling of label dependencies and enhances the coherence of predicted sequences. However, this model structure is relatively simplistic, predominantly concatenating individual sub-models. This leads to suboptimal overall model efficiency and may fail to fully capture crucial text features. The experimental results of this study indicate that simply stacking sub-models does not necessarily yield accuracy improvements (*Shen, Lin & Huang, 2016*). For instance, the inclusion of LSTM between BERT and CRF could inadvertently blur the representation of text features that BERT has processed, potentially impacting overall performance.

To better address these challenges and accurately identify named entities in the field of TCM, we propose Dyn-Att Net. This model optimizes the structure of the traditional BERT-BiLSTM-CRF model by employing parallel pathways for encoding and feature extraction of the input text sequence. One pathway utilizes BERT to encode words into word vectors and extract semantic features, while the other pathway employs LSTM to

capture contextual relationships. Following feature extraction, we introduce a dynamic attention mechanism to fuse the two representations based on their importance for TCM NER tasks, resulting in more effective text representation. Finally, the fused text representations are fed into the CRF layer for label prediction, effectively recognizing complex professional terminologies in TCM NER.

The main contributions of this article can be summarized as follows: (1) We introduce the Dyn-Att Net model for Traditional Chinese Medicine Named Entity Recognition (TCM NER) tasks. By incorporating a dynamic attention mechanism and fusing the representations from the BERT and LSTM pathways based on their importance, we achieve more effective text representations. (2) We propose a parallel architecture instead of a sequential structure to minimize unnecessary waiting time in TCM NER deep learning models. (3) Extensive experimental results demonstrate that our proposed Dyn-Att Net model can significantly improve the TCM NER performance. Furthermore, these results confirm our model's strong generalization capabilities and outstanding performance across various domains, as demonstrated through experiments on the APTNER, MSRA, and EduNER datasets.

The remainder of the article is organized as follows. 'Related Work' reviews the related work on TCM NER. 'Materials and Methods' presents the main idea of the proposed Dyn-Att Net model. 'Experimental Settings' demonstrates the experimental results and analysis. 'Experimental Results and Analysis' concludes our work.

# RELATED WORK

## Named entity recognition

The dictionary and rule based method is an early approach to NER that uses pre-built dictionaries and some rules to identify named entities in text (*Salah et al., 2022*; *Tarmizi & Saad, 2022*). *Humphreys et al. (1998)* developed the LaSIE-II system for the MUC-7 task, which used rule-based methods to implement a NER system. *Bao, Song & Zhang (2022)* proposed a novel NER method for Traditional Chinese Medical classics, combining semi-supervised learning and rule-based approaches. Experimental results demonstrated its effectiveness. However, this approach required much workforce to build dictionaries and rules, and identifying newly named entities may have been significantly affected.

The machine learning based method can better adapt to different text data types, fields, and languages than the rule-based method. Machine learning based methods for NER mainly include the hidden Markov model (HMM) (*Rabiner, 1989*), the support vector machines (SVM), and the conditional random fields (CRF) (*Lafferty, McCallum & Pereira, 2001*). Among them, the CRF model is the most common. *Lei et al. (2014)* used this model for NER in the medical records of the Peking Union Medical College Hospital, and the results were better than those of SVM and maximum entropy (ME). *Zhang et al. (2023)* stressed the need for multi-class NER in TCM, covering various entity types like herbal names, prescriptions, and medical conditions. Conventional machine learning is less suitable for TCM NER's complexity (*Xu et al., 2021*), so researchers combine domain knowledge and deep learning techniques to improve recognition, given the unique nature of TCM. *Wang et al. (2014)* concentrated on symptom name recognition (SNR) within

TCM records. They utilized sequence labeling techniques, favoring CRF over HMM and MEMM for SNR. However, the study's scope was restricted to SNR within chief complaints, which was one aspect of TCM records.

With the advancement of deep learning research, further optimizations have been made to NER. Deep learning enables the learning of more complex language features and handles problems in NER such as ambiguity and cross-domain recognition. Currently, deep learning-based methods for NER mainly include Word2Vec (*Mikolov et al., 2013b*), LSTM (*Greff et al., 2016*), Transformer (*Vaswani et al., 2017*), Bert (*Devlin et al., 2018*), and other models. *Deng, Fu & Chen (2021)* proposed a Robustly optimized Bidirectional Encoder Representations from Transformers a method for automatically recognizing entities in TCM patent texts, achieving superior performance over baseline methods. However, this method cannot handle complex sentence structures and specific domain terms in TCM patent texts. *Wang et al. (2022a)* proposed the BERT-BiGRU model and used Softmax to identify patients' diseases, thus helping TCM practitioners make clinical decisions. *Yu et al. (2022)* focused on Mineral NER from unstructured Chinese mineral texts, whereby significant achievements were obtained using the BERT-BiGRU-CRF model. *Chang et al. (2021)* proposed that the BERT-BiLSTM-IDCNN-CRF model enhances NER by addressing polysemy and context issues in TCM NER. Experiments on the CLUENER dataset showed 81.18% F1 score, outperforming the BiLSTM-CRF benchmark by 4.79%. *Yanling et al. (2021)* used BiLSTM and CRF to extract information from unstructured Chinese medicine records and built a Chinese medicine knowledge graph using Neo4j. *Souza, Nogueira & Lotufo (2019)* used the BERT-CRF architecture for Portuguese NER tasks and achieved good results by fine tuning the model on the HAREM I dataset.

The BERT-LSTM-CRF model is one of the most popular NER models and has been proven effective in many experiments. *Qu et al. (2020)* used the BERT-BiLSTM-CRF model for NER in TCM texts and showed excellent performance, partially solving the challenge of recognizing ambiguous entities in TCM. *Liu et al. (2023b)* addressed the need for structured knowledge in citrus pests and diseases, and proposed a model using BERT-BiLSTM-CRF for entity extraction. *Xuefeng et al. (2022)* developed an ALBERT-BiLSTM-CRF model for NER in TCM traumatology electronic medical records, which has fewer parameters and reduces hardware requirements during model training. *Yang et al. (2022)* compared four different pre-training models namely BERT, A Lite Bidirectional Encoder Representations from Transformers (ALBERT) (*Lan et al., 2019*), Robustly optimized Bidirectional Encoder Representations from Transformers approach (RoBERTa) (*Liu et al., 2019*), GPT2 (*Radford et al., 2019*), and GPT3 (*Brown et al., 2020*) using a TCM dataset. They confirmed the effectiveness of combining pre-training models with BiLSTM-CRF.

While the above mentioned research methods have shown promising results, they all adopt conventional sequential structures, leading to low efficiency and inability to effectively exploit the specific advantages of each algorithm and model. Hence, we propose a solution to this problem based on the dynamic attention network (Dyn-Att Net), which aims to overcome these limitations by introducing a more adaptive and dynamic approach, allowing for a more efficient integration of diverse algorithms and models. This, we believe,

will lead to a more effective and nuanced understanding of the intricate patterns present in TCM data.

## Attention mechanisms

The utilization of attention mechanisms in NER processes is paramount for enhancing the performance and efficacy of neural network-based models. By dynamically learning the importance weights of different positions in text sequences (*Galassi, Lippi & Torroni, 2020*), attention mechanisms facilitate the extraction of relevant information, thereby improving the accuracy and robustness of NER systems. Presently, a wealth of research substantiates the notion that the utilization of attention mechanisms typically yields commendable outcomes (*Gkoumas et al., 2021*; *Zhu et al., 2023*).

Attention mechanisms can enhance information utilization, as demonstrated by *Jin et al. (2019)*, *Zhang et al. (2022)*, *Liu et al. (2021)*, *Kong et al. (2021)*, and *Zhao et al. (2021)*, thus enabling models to capture both local and global information within text sequences effectively. This enhanced information utilization allows the models to discern subtle patterns and dependencies crucial for accurate entity recognition.

Models equipped with attention mechanisms, such as the character-based gated convolutional recurrent neural network with attention (GCRA) proposed by *Jin et al. (2019)*, and the Dynamic Cross and Self-lattice Attention Network (DCSAN) introduced by *Zhao et al. (2021)*, exhibit adaptability to complex textual structures. This adaptability is essential, particularly in languages like Chinese, where characters and words may convey nuanced semantic information.

Incorporating attention mechanisms allows for the integration of multi-level features, as demonstrated by *Kong et al. (2021)*. By combining multi-level convolutional neural networks (CNN) with attention mechanisms, the model can capture hierarchical representations of entities, enhancing the discernment of fine-grained correlations in the character-word space.

Attention mechanisms also have improved performance in domain-specific NER tasks, such as traditional Chinese medicine named entity recognition (*Liu et al., 2021*). By introducing novel word character-integrated self-attention modules, models can effectively identify domain-specific entities, showcasing the significance of attention mechanisms in addressing the challenges inherent to specialized domains.

In essence, the integration of attention mechanisms in NER processes enhances information utilization and adaptability, facilitates the integration of multi-level features, and improves performance in domain-specific tasks. Inspired by the aforementioned research methodologies, we designed a dynamic attention mechanism to effectively integrate the representations generated by the BERT and LSTM models. This mechanism adaptively adjusts the importance of BERT and LSTM models for the NER task, thereby fully leveraging the advantages of both models.

# MATERIALS AND METHODS

In this section, we commence by presenting an overview of the overall architecture of the Dyn-Att Net model. Following that, we provide a detailed explanation of the deep learning models employed within the Dyn-Att Net model. Finally, we elaborate on the deployment of the dynamic attention mechanism within the Dyn-Att Net model.

## Overall architecture

Due to the simple structure of the BERT-BiLSTM-CRF model and its low training efficiency, it takes work to handle more complex tasks. In this study, we propose the Dyn-Att Net model based on the traditional BERT-BiLSTM-CRF model. First, the input text sequence is not simply encoded into word vectors through BERT, but rather encoded and feature extracted through two different paths. One path uses BERT to encode words into word vectors and extract semantic features, and BERT itself encodes the text sequence into word vectors. The other path uses LSTM to extract contextual relationships. Since LSTM cannot encode words into word vectors, we first use Word2Vec to build a word vector table for the corresponding corpus, encode the words in the input sequence into word vectors by looking up the word vector table, and then pass them to LSTM. For the features extracted by these two paths respectively, this study uses a dynamic attention mechanism to fuse them, which is conducive for retaining the advantages of both the BERT and LSTM models. Next, the fused attention value matrix is passed into CRF as an emission matrix, combined with the transition matrix trained by CRF itself, and the label with the highest probability is selected as the output of the final model. The Dyn-Att Net model is shown in Fig. 2.

## Dyn-Att net layer

After processing the input text through parallel BERT and Word2Vec-LSTM models, two distinct feature representations are obtained. The BERT-based representation encompasses richer semantic features, while the Word2Vec-LSTM-based representation captures more contextual features. Subsequently, these two sets of features are passed to the Dyn-Att net layer, where they complement each other's advantages, combining to yield a higher-level feature correlation representation. The Dyn-Att net is a type of attention mechanism used in deep learning, which can help models automatically learn higher-level feature representation in different feature representations and dynamically adjust weights to improve performance. For NER tasks, the attention weights of context information and semantic relationships can be dynamically adjusted to improve the accuracy of NER. This study designs a dynamic attention structure based on this principle. The model first averages and transforms the outputs of LSTM and BERT into two numerical values using a linear network. Then, the *tanh* activation function is used to map these two values into the range of $[-1,1]$, and they are concatenated into a 2D vector. The attention weight values are computed using the *Softmax* function, and the outputs of LSTM and BERT are weighted accordingly. Finally, the weighted results are added to obtain the attention values dynamically combining BERT and LSTM. This approach can help models automatically learn critical information in input sequences and dynamically adjust weights to improve

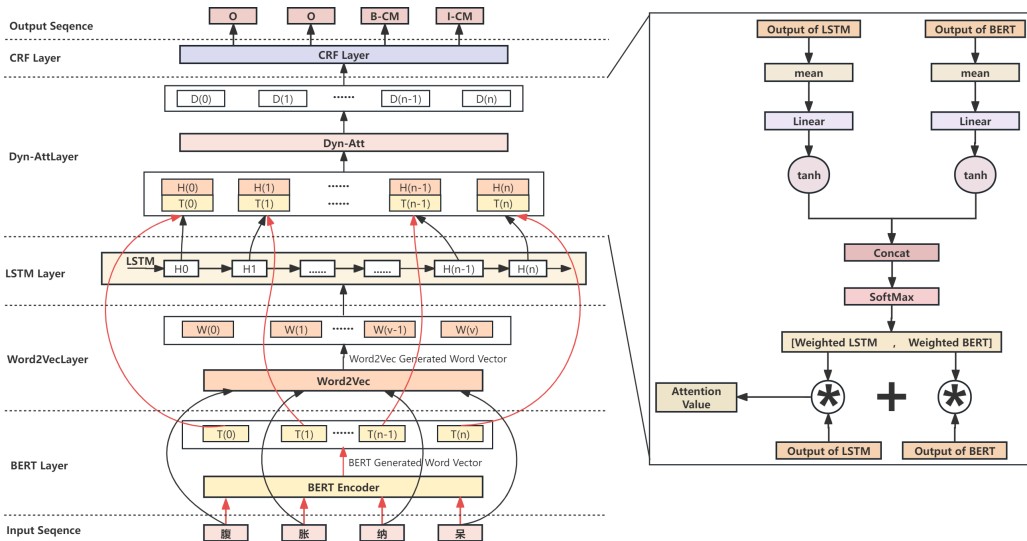

**Figure 2** The overall architecture of the Dyn-Att Net.

performance, thereby improving model robustness and optimizing model structure. The Dyn-Att Net structure is shown in the right side of Fig. 2.

We denote the outputs processed through LSTM as $T$ and those processed through BERT as $B$. First, we compute the mean of these two vectors, followed by linear transformation and activation function processing:

$T' = mean(T)$

$B' = mean(B)$

$T'' = Linear(T') = W_{t'}T' + b_{t'}$

$B'' = Linear(B') = W_{b'}B' + b_{b'}$

$T''' = tanh(T'')$

$B''' = tanh(B'')$.

Where $W_{t'}$, $W_{b'}$ are weights, $b_{t'}$ and $b_{b'}$ are bias. Next, by concatenating the two vectors and mapping them to the same space, and then applying a *Softmax* operation, we obtain the corresponding weight coefficients for $T$ and $B$. These weight coefficients dynamically adjust with variations in input and the effectiveness of LSTM and BERT:

$Concat(T''', B''') = [T'''; B''']$

$[W_t; W_b] = Softmax(Concat(T''', B'''))$

Finally, the attention values, denoted as $V$, are computed by combining the results:

$V = W_t T + W_b B$.

## BERT layer

BERT (*Devlin et al., 2018*) is an encoder model based on the Transformer architecture. By pre-training on a large corpus of unannotated data, it learns a universal language representation, which can be fine-tuned for downstream tasks such as text classification, sequence labeling, and question answering. The word embedding layer of BERT includes three parts: token embedding, segment embedding, and position embedding, which are

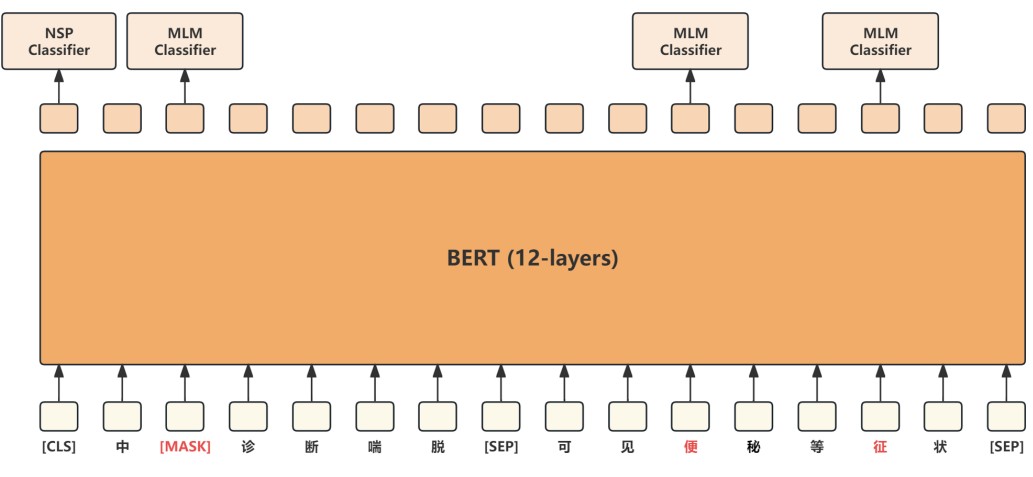

**Figure 3** BERT model.

summed together to represent the input sequence as a vector. Pre-training in BERT consists of two main phases: masked language model (MLM) and next sentence prediction (NSP), which learn language modeling and sentence relationships. MLM requires BERT to predict masked words, like completing a fill-in-the-blank exercise, while NSP randomly selects two sentences and predicts whether they are contiguous. BERT models generally come in 12 layers and 24 layers, with each layer composed of Transformers. Their main contribution is the universal language representation learned through pre-training, which can be shared and transferred across various natural language processing tasks, significantly improving their performance. The BERT model is shown in Fig. 3.

## Word2Vec layer

Word2Vec (*Mikolov et al., 2013a*) is a type of distributed representation model for word vectors. It includes two models, continous bag-of-word (CBOW) and Skip-gram. In the CBOW model, the center word is inferred from the words in its context, which is defined by the window size. The inference process uses a three layer neural network, so the text must be onehot encoded before being passed into the CBOW model to vectorize the words for computation and optimization. Finally, *Softmax* is used to calculate the probability and output the word with the highest probability, achieving the goal of predicting the center word. The Skip-gram model is the opposite of the CBOW model, inferring the context words from the center word. The CBOW and Skip-gram models are shown in Fig. 4.

## LSTM layer

LSTM (*Hochreiter & Schmidhuber, 1997*) is a type of recurrent neural network (RNN) model developed to address the issue of vanishing and exploding gradients in traditional RNN models. In traditional RNNs, the gradient can exponentially increase or decrease during backpropagation, making it difficult to propagate the gradient effectively in deep networks. LSTM overcomes this problem by introducing structures such as forget gates,

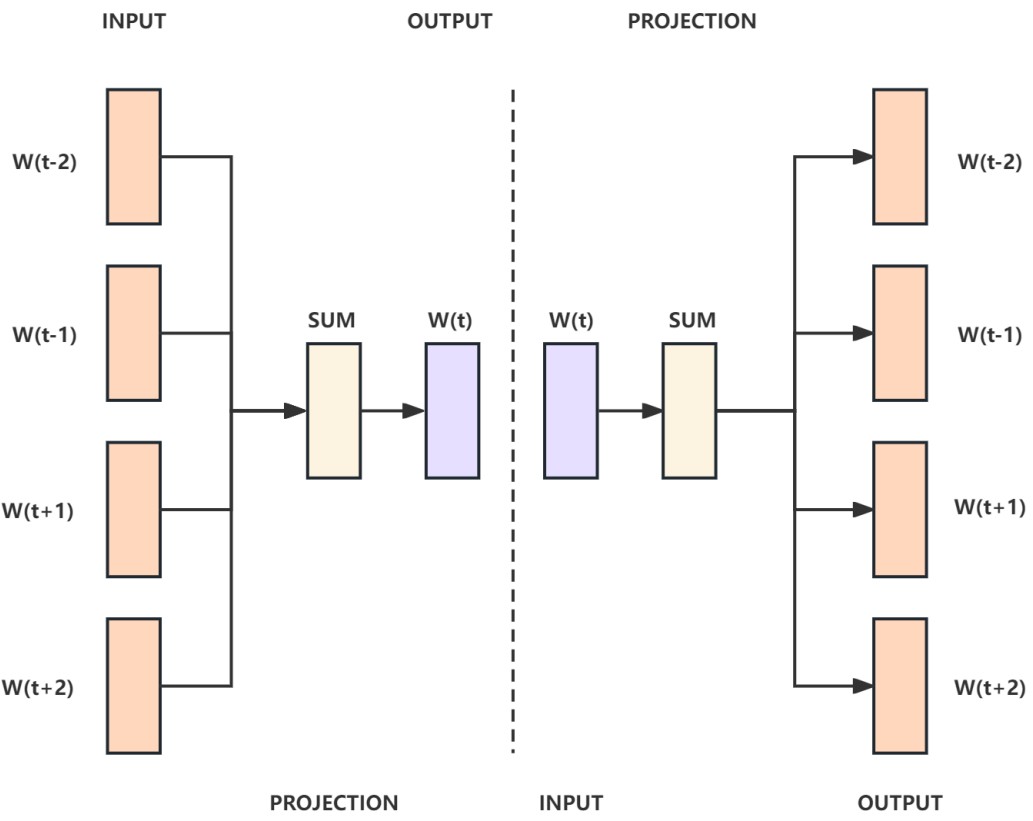

INPUT  OUTPUT  PROJECTION

W(t-2)

W(t-1)  SUM  W(t)  W(t)  SUM  W(t-2)

W(t+1)  W(t-1)

W(t+2)  W(t+1)

PROJECTION  INPUT  W(t+2)  OUTPUT

**Figure 4** The CBOW model on the left and the Skip-gram model on the right.

update gates, and output gates to control the gradient, and it has achieved good results in long sequence tasks. The architecture of LSTM is shown in Fig. 5.

Firstly, there are three inputs to the LSTM model: $C_{t-1}$, which records information from the previous time step, and $H_{t-1}$, which represents the hidden state. Furthermore, $X_t$ represents the input information at the current time step. $H_{t-1}$ and $X_t$ are concatenated and passed to the LSTM model. The first gate is the forget gate, which uses the *sigmoid* function to forget the less important information. The second gate is the update gate, split into two paths: one path normalizes the input using the *tanh* activation function. In contrast, using the *sigmoid* function, the other path forgets the less critical information. The final result is obtained by multiplying the results of the two paths and adding them to $C_{t-1}$ to obtain the updated information $C_t$, which will be the input for the next time step. The third gate is the output gate, which applies the *tanh* function to normalize $C_t$ and uses the *sigmoid* function to determine the information to be output for the current time step.

$$F_t = \delta(W_f X_t + U_f H_{t-1} + b_f)$$
$$I_t = \delta(W_i X_t + U_i H_{t-1} + b_i)$$
$$O_t = \delta(W_o X_t + U_o H_{t-1} + b_o)$$
$$\hat{C}_t = tanh(W_c X_t + U_c H_{t-1} + b_c).$$

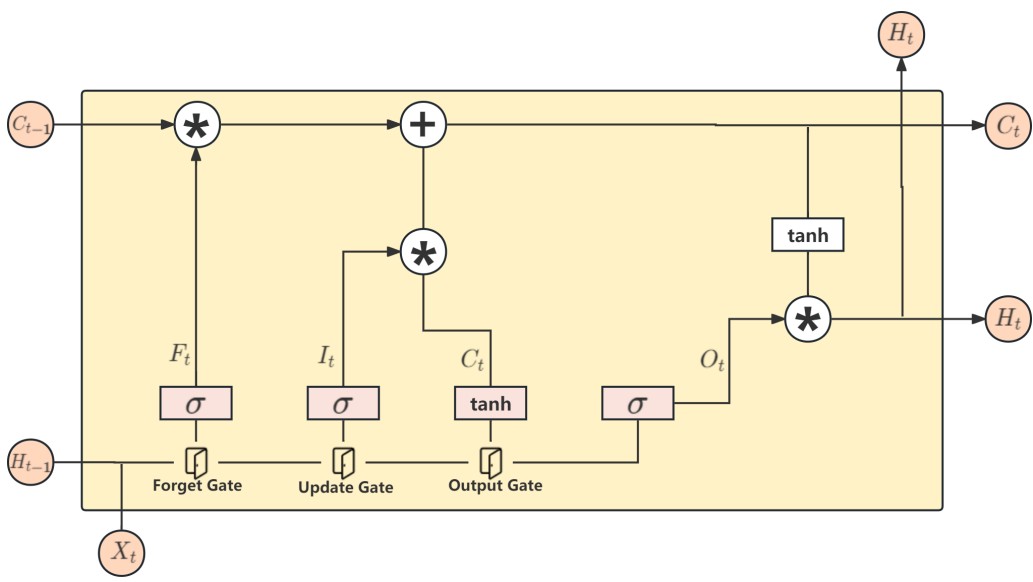

**Figure 5  LSTM model.**

The above explanation outlines the process of computing. The specific formulas for these calculations are shown below, where $W$ and $U$ are weight values, $b$ is the bias parameter, and $\delta$ represents the *sigmoid* function.

## CRF layer

CRF (*Sutton & McCallum, 2012*) is an undirected graphical model in which nodes represent variables and edges represent relationships between variables, and the weights on nodes and edges are all related to probabilities. The CRF model predicts the optimal label sequence through Viterbi decoding, implemented based on dynamic programming. The expression for Viterbi decoding is as follows:

$$y^* = argmax \quad p(y|x^{'}W, b).$$

Due to the constraints set on the predicted labels by CRF, the model can avoid illegal label sequences and improve recognition accuracy. Suppose a text input sequence is $X = (X_1 + X_2 + .. + X_i + .. + X_n)$, where the subscript of $X$ represents the index position of the word in the text. The corresponding output sequence to the text is $Y = (Y_1 + Y_2 + .. + Y_i + .. + Y_n)$. When the sequence satisfies the corresponding Markov distribution, it can be represented by the CRF model:

$$p(y|x; W, b) = \frac{\prod_{i=1}^{n} \phi_i(y_{i-1}, y_i, x)}{\sum_{y' \in Y(z)} \prod_{i=1}^{n} \phi_i(y'_{i-1}, y'_i, x)}.$$

Here, $\phi_i(y'_{i-1}, y'_i, x) = EXP(W_{y',y}^T x_i + b_{y_i,y})$. In this formula, $W_{y',y}^T$ represents the weight parameter of predicting the next label $y$ when the previous label is $y'$, and $b_{y_i,y}$ represents the offset of this linear expression. The numerator represents the probability of the optimal path, and the denominator represents the probability of all possible paths.

The calculation method of the loss function in CRF is different, and it usually uses the log-likelihood function for calculation. The expression for the log-likelihood function is as follows:

$$L(W, b) = \sum_i \log p(y|x; W, b).$$

# EXPERIMENTAL SETTINGS

To validate the effectiveness of the Dyn-Att Net model, this topic encompasses a comprehensive experimental design. We begin by introducing the datasets employed in this research. Subsequently, we present the four evaluation metrics utilized in the experiments. Finally, we provide details regarding the experimental setup, including the environmental conditions and model parameters.

## DataSet

The dataset (*Qiangchuan, 2022*) used in this study was sourced from the open source dataset provided by Baidu on the PaddlePaddle platform. The dataset consists of 6,574 sentences, with approximately 5,259 sentences in the training set, 657 sentences in the validation set, and 658 sentences in the test set. The dataset primarily consists of electronic medical records in traditional Chinese medicine. It encompasses 10 labels, including 15,846 entity types and 183,900 non-entity types. The details are shown in Table 2.

BIO tags are used in this study dataset. The primary role of BIO tags is to label whether a word belongs to an entity and provide information for each entity's start and end position. Due to the specificity of text data in the field of TCM, it contains a considerable number of entity types; there are are 11 entity types used in this study. The BIO annotation form is shown in Table 3.

## Evaluation indices

In NER tasks, the following four metrics are commonly used to evaluate the performance of models. Precision refers to the proportion of samples predicted by the model as named entities. Accuracy refers to the ratio of all correctly predicted samples to total samples. Recall refers to the proportion of all indeed named entities correctly expected as named entities by the model. The F1-score is the weighted average of precision and recall, representing a performance metric considering both precision and recall. The corresponding calculation formula is as follows:

$$Precision = \frac{TP}{TP+FP} \times 100\%$$
$$Accuracy = \frac{TP+NP}{TP+FN+FP+TN} \times 100\%$$
$$Recall = \frac{TP}{TP+FN} \times 100\%$$
$$F1-score = \frac{2 \times Precision \times Recall}{Precision+Recall} \times 100\%.$$

## Experimental environment

The experimental environment is shown in Table 4.

## Experimental parameters

*Isnain, Sihabuddin & Suyanto (2020)* employed Word2Vec with a CBOW structure for tweet vectorization, followed by LSTM for feature extraction to detect hate speech. They

**Table 2  Distribution of entity types in the dataset.**

| Entity category name | Number |
|---|---|
| TCM diagnosis | 336 |
| Western medicine diagnosis | 3,082 |
| TCM treatment | 1,325 |
| Chinese herbs | 3,540 |
| TCM syndrome | 1,464 |
| Prescription | 1,243 |
| TCM treatment principles | 397 |
| Clinical manifestations | 3,812 |
| Western medicine treatment | 562 |
| Other treatments | 85 |

**Table 3  BIO tagging meaning.**

| BIO | Marker meaning |
|---|---|
| B | Indicates the location of the start of the entity |
| I | Indicates the position inside the entity |
| O | A tag indicating that it does not belong to any entity |

**Table 4  Experimental environment.**

| Experimental environment configuration | Introduce |
|---|---|
| Deep learning framework | Pytorch 1.10.1 |
| Programming language | Python 3.8 |
| Memory | 8GB |
| Hard disk | 1TB |
| GPU | RTX 3090 |

achieved a commendable F1 score of 96.29%. Their CBOW training involved 10 epochs, a hidden layer with 200 neurons, and a window size of 5. Consequently, we adopt these identical parameters for the Word2Vec layer of the Dyn-Att Net model in our study, aiming for improved outcomes. Table 5 provides the specific parameter settings.

In order to find a set of parameters suitable for the Dyn-Att Net model, the experiments in this subsection focus on four parameters of the model namely Epoch, Batch size, optimizer, and learning rate. The results of the four comparison experiments are presented in Tables 6 to 9.

After conducting various experiments and evaluations, we determined the optimal parameter settings for our model. In the Epoch parameter comparison experiment, increasing the Epoch value resulted in improved accuracy, precision, recall, and F1 score. The model exhibited its best performance at Epoch 30 and 35, with identical metrics. This result suggests that increasing the epoch beyond 30 did not lead to further improvements in model performance. Therefore, we chose Epoch 30 as the optimal choice. Similarly, in the batch size comparison experiment, a batch size of 64 yielded the highest accuracy,

**Table 5  Word2vec algorithm related parameters.**

| Parameter names within the Word2Vec | Parameter value |
|---|---|
| Epoch | 10 |
| Vector size | 200 |
| Window size | 5 |
| Word vector model | CBOW |

**Table 6  Epoch parameter comparison experiment.**

| Epoch | Precision | Recall | F1 score |
|---|---|---|---|
| 10 | 78.99% | 74.53% | 76.53% |
| 20 | 78.77% | 77.19% | 77.97% |
| 30 | 80.26% | 83.76% | 81.91% |
| 35 | 80.26% | 83.76% | 81.91% |

**Table 7  Batch size parameter comparison experiment.**

| Batch size | Precision | Recall | F1 score |
|---|---|---|---|
| 32 | 80.99% | 77.58% | 79.24% |
| 64 | 80.26% | 83.76% | 81.91% |
| 128 | 77.96% | 77.05% | 77.41% |

precision, recall, and F1 score, making it the preferred choice. Furthermore, we explored different optimizers (SGD, ADAM, and ADAMW) and found that the ADAMW optimizer delivered the best performance in terms of accuracy, precision, recall, and F1 score. Lastly, we investigated various learning rates (2e−4, 2e−5, and 2e−6) and concluded that a learning rate of 2e−5 achieved the highest metrics, with lower and higher rates exhibiting suboptimal performance. Therefore, our parameter comparison experiments identified optimal parameter settings, leading to improved model performance across various metrics, as depicted in Table 10.

# EXPERIMENTAL RESULTS AND ANALYSIS

In this section, we first verify the effectiveness of the Dyn-Att Net model for TCM NER and its capability to optimize the traditional BERT-LSTM-CRF model through the comparison of different models and comparison with the basic model. Subsequently, we experiment with three NER datasets: APTNER, MSRA, and EduNER, derived from different domains, to demonstrate the commendable generalization capability of the Dyn-Att Net model.

## Comparison of different models

We evaluated the performance of classical models within the field of TCM on our dataset, using it as a benchmark for assessing the Dyn-Att Net model. A comprehensive comparison of all model evaluations is presented in Table 11. The fluctuation of accuracy and loss values for the Dyn-Att Net model is depicted in Fig. 6.

**Table 8 Optimizer parameter comparison experiment.**

| Optimizer | Precision | Recall | F1 score |
| --- | --- | --- | --- |
| SGD | 80.60% | 73.43% | 76.84% |
| ADAM | 79.43% | 74.61% | 76.91% |
| ADAMW | 80.26% | 83.76% | 81.91% |

**Table 9 Learning rate parameter comparison experiment.**

| Learning rate | Precision | Recall | F1 score |
| --- | --- | --- | --- |
| 2e−4 | 76.03% | 57.64% | 65.57% |
| 2e−5 | 80.26% | 83.76% | 81.91% |
| 2e−6 | 71.18% | 63.28% | 66.95% |

We compared models 6, 7, 8, 9 with models 1 and 2, as illustrated in Table 11, and observed a significant performance boost when employing BERT. This underscores the vital role of BERT in enhancing overall NER task performance. Furthermore, an overview of the entire experimental model revealed that the addition of CRF or LSTM layer indeed improved model performance to some extent. This serves as evidence that within complex domains like TCM, CRF's sequence labeling and label transition optimization, as well as LSTM's context understanding, contribute to model effectiveness. However, it is noteworthy that in the comparison of models 7, 8, and 9, we noticed that the actual performance either did not reach the theoretical expectations or, in some cases, decreased when adding CRF to BERT-LSTM or LSTM to BERT-CRF models. This indicates that blindly adding algorithms to traditional model structure increases the complexity of the entire model, leading to unstable or even decreased model recognition.

To optimize the model structure, this study introduces the Dyn-Att Net model, which enhances traditional NER models by incorporating a dynamic attention mechanism. The Dyn-Att Net model is implemented in two variants, one based on BERT and the other on ALBERT. In our study, when evaluating these BERT and ALBERT-based models, we found that BERT consistently outperforms ALBERT across various metrics such as accuracy, precision, recall, and F1 score in the TCM NER task we examined.

## Comparison with the basic model

As shown in Table 11, the Dyn-Att Net consistently enhances the overall model performance, whether integrated with ALBERT or BERT, thus highlighting its positive impact. Subsequently, due to BERT's superior performance, we concentrate primarily on the comparative analysis of BERT-based models to affirm the effectiveness of the BERT-based Dyn-Att Net model.

Firstly, compared to the widely-used BERT-LSTM-CRF model, our optimized model has significantly improved all performance indicators, with the F1 score increasing from 80.69% to 81.91%. Consequently, the BERT-based Dyn-Att Net model demonstrates notable performance enhancements over the BERT-LSTM-CRF model. Additionally, the BERT-based Dyn-Att Net model outperformed the SoftLexicon model (*Ma et al., 2020*)

**Table 10  Dyn-Att net model parameters.**

| Model parameter name | Parameter value |
|---|---|
| Epoch | 30 |
| Learning rate | 2e−5 |
| Batch size | 64 |
| Max len | 128 |
| Lstm units | 128 |
| Optimizer | AdamW |

**Table 11  Evaluation results of all models.** *Denotes results generated from our experiments.

| Model number | Model name | Accuracy | Precision | Recall | F1 score |
|---|---|---|---|---|---|
| 1 | LSTM* | 83.74% | 29.55% | 45.13% | 35.71% |
| 2 | LSTM+CRF* | 84.59% | 38.04% | 38.25% | 38.13% |
| 3 | SoftLexicon* | 91.34% | 78.19% | 82.96% | 80.46% |
| 4 | ALBERT+LSTM+CRF* | 88.22% | 70.84% | 62.04% | 66.14% |
| 5 | Dyn-Att Net based on ALBERT* | 89.63% | 76.73% | 71.21% | 73.92% |
| 6 | BERT* | 91.74% | 78.51% | 82.36% | 80.16% |
| 7 | BERT+LSTM* | 92.04% | 78.73% | 83.72% | 81.03% |
| 8 | BERT+CRF* | 91.59% | 79.86% | 82.52% | 81.32% |
| 9 | BERT+LSTM+CRF* | 91.25% | 78.04% | 83.64% | 80.69% |
| 10 | Dyn-Att Net based on BERT | 92.06% | 80.26% | 83.76% | 81.91% |

across all metrics, with a noteworthy improvement of 1.45% in F1 score. Furthermore, comparing the performance indicators of the BERT-based Dyn-Att Net model with models 6 and 7 demonstrates significant improvements across all four metrics, underscoring its superiority over using only the BERT model or the BERT-LSTM model. Finally, when comparing the BERT-based Dyn-Att Net model with models 7, 8, and 9, we observed that the optimization of the Dyn-Att Net structure indeed yielded certain benefits. It effectively leveraged each sub-model of model 9, resulting in consistent improvements across all metrics compared to models 7 and 8.

The comparison of the core model running time is shown in Table 5. It is noticeable that the BERT-based Dyn-Att Net model exhibits a shorter runtime compared to the BERT+LSTM+CRF model, which suggests that the optimized parallel structure is more efficient and reduces processing time.

Based on the above analysis, it is evident that the BERT-based Dyn-Att Net model not only enhances TCM NER performance but also optimizes traditional model structures to improve efficiency.

## Comparison with the APTNER dataset

To further validate the model's generalization capabilities in complex domains, we conducted training and evaluation on the cyber threat intelligence (CTI) dataset introduced by *Wang et al. (2022b)*. This dataset (*Wangxuren, 2022*), currently the largest in the CTI field, encompasses 21 distinct named entities, posing a substantial challenge. Our model's

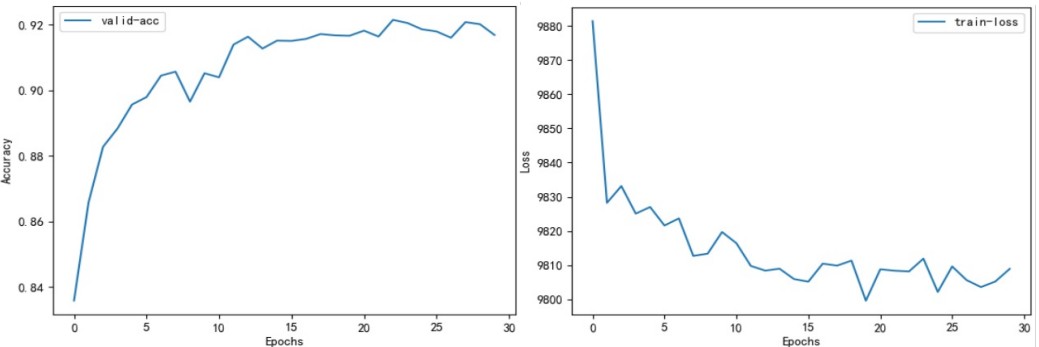

**Figure 6** On the left is the figure depicting changes in model accuracy, and on the right is the figure displaying fluctuations in loss.

**Table 12** Core model runtime.

| Model name | Running time |
| --- | --- |
| BERT+LSTM+CRF | 908.5041613578796 s |
| Dyn-Att Net based on BERT | 886.1882770061493 s |

**Table 13** Comparative experiment of the Dyn-Att Net model with the APTNER dataset. [1]Indicates results from *Wang et al. (2022b)*.

| Model number | Model name | Precision | Recall | F1 score |
| --- | --- | --- | --- | --- |
| 1 | BiLSTM-CRF[1] | 68.78% | 67.2% | 67.98% |
| 2 | CNN-BiLSTM-CRF[1] | 69.23% | 67.08% | 68.14% |
| 3 | LM-LSTM-CRF[1] | 70.77% | 65.46% | 68.01% |
| 4 | BiLSTM-CRF+ELMo[1] | 80.62% | 78.68% | 79.64% |
| 5 | BERT-BiLSTM-CRF [1] | 82.99% | 81.64% | 82.31% |
| 6 | Dyn-Att Net based on BERT(Our Model) | 89.79% | 90.88% | 90.33% |

performance in this domain proved exceptional, surpassing that of other models by a significant margin.

As illustrated in Table 13, when compared to the BERT-BiLSTM-CRF model, our model exhibited substantial improvements across various performance metrics. Specifically, our model achieved a 6.8% increase in precision, a 9.24% boost in recall, and a notable 8.02% enhancement in the F1 score. Meanwhile, compared to the BiLSTM-CRF (*Lample et al., 2016*), CNN-BiLSTM-CRF (*Ma & Hovy, 2016*), LM-LSTM-CRF (*Liu et al., 2018*), and BiLSTM-CRF+ELMo (*Peters et al., 2018*) models, the Dyn-Att Net model exhibits improvements across all performance metrics.

## Comparison with the MSRA dataset

Table 14 presents a comparative analysis of the Dyn-Att Net model on the MSRA dataset (*Levow, 2006*). The MSRA dataset is a commonly used dataset for Chinese NER, published by Microsoft Research Asia. It consists of Chinese texts from various sources such as news,

**Table 14** Comparative experiment of the Dyn-Att Net model with the MSRA dataset. [2]Indicates results from *Johnson, Shen & Liu (2020)*.

| Model number | Model name | Precision | Recall | F1 score |
|---|---|---|---|---|
| 1 | Conditional probabilistic models[2] | 91.22% | 81.71% | 86.2% |
| 2 | Multi-phase model[2] | 88.94% | 84.2% | 86.51% |
| 3 | Graph-based semi-supervised[2] | 90.62% | 77.84% | 83.74% |
| 4 | adversarial transfer learning[2] | 91.73% | 89.58% | 90.64% |
| 5 | Dyn-Att Net based on BERT (Our Model) | 92.83% | 89.66% | 91.21% |

**Table 15** Comparative experiment of the Dyn-Att Net model with the EduNER dataset. [3]Indicates results from *Li et al. (2023)*.

| Model number | Model name | Precision | Recall | F1 score |
|---|---|---|---|---|
| 1 | BiLSTM+CRF[3] | 71.74% | 51.90% | 60.23% |
| 2 | LR-CNN[3] | 64.87% | 60.15% | 62.42% |
| 3 | SoftLexicon[3] | 67.02% | 63.24% | 65.07% |
| 4 | Dyn-Att Net based on BERT (Our Model) | 71.77% | 64.03% | 67.63% |

blogs, and web articles, used for training and evaluating NER algorithms. The dataset is finely annotated with entity categories including personal names, location names, and organization names, assisting researchers in developing and evaluating Chinese named entity recognition models.

As observed in Table 14, our Dyn-Att Net model outperforms the other three models in terms of precision, recall, and F1 score. Specifically, it achieves a precision of 92.83%, recall of 89.66%, and F1 score of 91.21%. compared to the conditional probabilistic models (*Chen et al., 2006*), multi-phase model (*Zhou et al., 2006*), graph-based semi-supervised (*Han et al., 2015*), and adversarial transfer learning models (*Cao et al., 2018*), our Dyn-Att Net model demonstrates superior performance across all metrics.

## Comparison with the EduNER dataset

The EduNER (*Li et al., 2023*) dataset is a Chinese NER dataset focused on the education domain, meticulously curated from various sources such as textbooks, academic papers, and education-related web pages. The dataset defines an education-specific NER schema by domain experts and is annotated by trained annotators. EduNER comprises 16 entity types, including 11,000 sentences and 35,731 entities, making it the first publicly available dataset tailored for the education domain NER task.

Our comparative experiments with the Dyn-Att Net model with the EduNER dataset, as shown in Table 15, demonstrate that the Dyn-Att Net model outperforms the other three models across all metrics. Specifically, compared to the BiLSTM+CRF model, the Dyn-Att Net model achieves a 7.4% improvement in F1 score. Similarly, compared to the LR-CNN model (*Gui et al., 2019*), it achieves a 5.21% improvement, and compared to the SoftLexicon model (*Ma et al., 2020*), it achieves a 2.56% improvement in F1 score.

# CONCLUSION AND PROSPECT

This article addresses the structural simplicity of the traditional named entity recognition model. The Dyn-Att structure is proposed based on BERT-LSTM-CRF. There are two improvements of the Dyn-Att Net model compared to traditional NER models. The first improvement is that it uses parallel connections instead of sequential connections for deep learning models. The second improvement is the addition of a dynamic attention mechanism to combine the features extracted by BERT and LSTM. By dynamically adjusting the importance of the features processed by BERT and LSTM, the model structure is optimized and recognition accuracy is improved. Compared to the currently most popular BERT-LSTM-CRF model, the F1 score of this model is improved by 1.22%, and it consumes less time. Moreover, its strong generalization capability is confirmed through validation on the APTNER, MSRA, and EduNER datasets. Therefore, the Dyn-Att-Net model not only effectively recognizes named entities in TCM but also demonstrates commendable cross-domain generalization capability. Although the Dyn-Att Net model demonstrates promising performance, it faces the challenge of entity imbalance within the field of traditional Chinese medicine (TCM). For example, there are 336 entities related to TCM diagnosis and 3,540 entities related to Chinese herbs in TCM dataset, representing a nearly tenfold difference in the quantities of entities between these two categories. Currently, we lack strategies to address this imbalance in entity quantities. In the future, we can employ active learning techniques and data augmentation to mitigate this issue and reduce labeling costs.

## Funding

This work is supported by Universiti Kebangsaan Malaysia. The funders had no role in study design, data collection and analysis, decision to publish, or preparation of the manuscript.

## Grant Disclosures

The following grant information was disclosed by the authors:
Universiti Kebangsaan Malaysia.

## Competing Interests

The authors declare there are no competing interests.

## Author Contributions

- Jingming Hou conceived and designed the experiments, performed the experiments, analyzed the data, performed the computation work, prepared figures and/or tables, authored or reviewed drafts of the article, and approved the final draft.
- Saidah Saad performed the experiments, analyzed the data, authored or reviewed drafts of the article, and approved the final draft.
- Nazlia Omar analyzed the data, authored or reviewed drafts of the article, and approved the final draft.

## Data Availability

The upload relevant codes, and APTNER dataset are available at GitHub: https://github.com/wangxuren/APTNER.

The MSRA dataset is available at: https://www.modelscope.cn/datasets/iic/msra_ner/files.

The EduNER dataset is available at GitHub: https://github.com/anonymous-xl/eduner.

Dyn-Att Net is available at Zenodo: Hou, J., Saad, S., & Omar, N. (2024). Dyn-Att Net. Zenodo. https://doi.org/10.5281/zenodo.10876525

The raw data and code are available in the Supplemental Files.

## Supplemental Information

Supplemental information for this article can be found online at http://dx.doi.org/10.7717/peerj-cs.2022#supplemental-information.

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
