# Peer review of "Enhancing traditional Chinese medical named entity recognition with Dyn-Att Net: a dynamic attention approach"

_PeerJ Computer Science, doi:10.7717/peerj-cs.2022_

## Round 0.1 · original submission · Major Revisions

Thanks for your submission to PeerJ Computer Science. I'm glad that the reviewers are interested in your work. However, they also proposed some issues. Please revise the work accordingly.

**Language Note:** PeerJ staff have identified that the English language needs to be improved. When you prepare your next revision, please either (i) have a colleague who is proficient in English and familiar with the subject matter review your manuscript, or (ii) contact a professional editing service to review your manuscript. PeerJ can provide language editing services - you can contact us at [email protected] for pricing (be sure to provide your manuscript number and title). – PeerJ Staff

·

Basic reporting

Abstract:
- The abstract is already in a good structure. If there is something that i want you to add is a one
sentence that declaring the hypothesis (what do you want to prove?) of your research. In other words,
what will happen if your approach is implemented compared to previous research?

Introduction:
- Your introduction needs more explanation on how NER can be implemented on identifying the specific meaning on your dataset/TCM dataset in general.
- I suggest you add several sentence on the general technical part on how the NER will work, and also a justification on why combining BERT and LSTM.
- I also suggest you to add a brief explanation of attention mechanism and dynamic strategy before the contributions part at Lines 75-76 to justify your approach.

General:
- There are some minor misspelled words, for example (bet-ween, under-standing, ma-chines, relation-ship, etc)

Experimental design

Experiment & Result:
- In Table 6, you conclude that the optimum epoch is 30. I suggest you add 1 more epoch more than 30 that will justify your result. So that the reader will know the optimum epoch is 30.
- It goes along with other parameters, batch size.
- Table 7 already correct, you can look that for reference.
- I suggest you add one conclution details at the ending to justify your result and conclude the paragraph on line 284-285.
- The comparison section is very nice and very informative. If there is something that i want you to add is you can cite and compare also with other research from your related work
that used either same approach or dataset.
- For example on the Table 12. Is the model BERT+LSTM+CRF is also done by you or other researcher? You need to justify that as well.
If not, you can cite the corresponding paper. If yes, you can add 1 other research that related to your research as i mentioned on previous point.

Conclusion:
- Please re-construct the sentences from line 347-250. You can simplify the sentences that it can be read more clearly.

Validity of the findings

- Your experiment result is very promising that I believe in the future especially on TCM domain there will be more research and experiment with other approach.

- In addition, I suggest you find other previous experiment on this matter and add to your next paper.
If the available paper/articles is very outdated, you can still include it and make the justification of it.

Additional comments

Overall the paper is very constructed and well balanced on the technical part. The improvement that can be made is on the justification of your findings and novelty of your work with other research that used same domain. Other than that is the representation of your result can be more detailed that i already discuss on the review.

Reviewer 2 ·

Basic reporting

The manuscript "Enhancing Traditional Chinese Medical Named Entity Recognition with Dyn-Att Net: A Dynamic Attention Approach" presents a novel approach for TCM NER task. It aims to improve upon the limitations of existing BERT-LSTM-CRF models by integrating dynamic attention mechanisms. The paper is well-structured, with clear language and relevant literature cited. The structure of the manuscript conforms to PeerJ standards and discipline norm. The manuscript provides the raw data.

To further improve the submission, I suggest:
1) There are some description errors in the manuscript. For example, the mention of "multimodal sentiment analysis performance" in line 83 may be incorrect.

2) There are some errors in the symbols , such as the symbols used for "next label" and "previous label" in line 238, which may need revision.

3)There is inconsistency in terminology throughout the manuscript. For example, "distinct categories" in line 64, "entity types" in line 252 and "entity categories" in line 256.

4)There are spelling errors in words,for example, lines 113, 116, 155, and 180.

5)It is preferable to have citations for conclusions drawn from previous work, such as the possible performance degradation mentioned in lines 73-74; the claim made in lines 66-67 that ' the TCM domain is replete with specialized terminologies, contributing to increased structural complexity within the text'; and the statement in line 104 that 'Conventional machine learning is less suitable for TCM NER’s complexity'.

6)In the introduction section, after discussing the challenges of TCM NER and before presenting the contributions of this paper, it is suggested to add descrption about the idea of the manuscript to address these challenges.

7)The parallel architecture is an important contribution of this manuscript. It is suggested to add the technical details in the manuscript.

8)In the related work section, it is suggested to include research related to attention mechanisms.

9)In the model description section, it is suggested to add citations for the Bert layer, word2Vec layer, LSTM layer, and CRF layer.

10) The figures listed in the manuscript are relevant and appropriately labeled and described, but the clarity of the images needs improvement. For example, the clarity of figure 2 and figure 3 is insufficient. Figure 7 is difficult to read due to the similar colors .

Experimental design

The manuscript compared the Dyn-Att Net model primarily with the BERT-LSTM-CRF model. The experimental results demonstrate the effectiveness of the proposed model. Moreover, the results on the APTNER dataset indicate that the model possesses cross-domain generalization capabilities.

To further improve the submission, I suggest:
1)In lines 296-230 of the manuscript, the author mentions "However, it's noteworthy that in.... unstable or even decreased model recognition." In Table 11, AlBert+LSTM+CRF is better than ALBERT+LSTM and ALBERT+CRF. It appears to be contrary to the previous discussion.

2) The current comparison experiments are inadequate, and it is recommended to include additional baseline models in the comparative analysis to ensure a comprehensive evaluation.While the results demonstrate the model's effectiveness, a more detailed comparison with state-of-the-art models could highlight its novelty and impact.

3) In the section of comparison on the APTNER dataset, the author used cyber threat intelligence data sets and achieved good results in comparison with traditional models to discuss the cross-domain capabilities of the model. The results of experiments on only one data set are insufficient. It is suggested to add more domain data sets for comparative experiments.

4) Please check the F1 score of each model in Table 11 of the manuscript. Using the F1score calculation formula provided in the manuscript, the calculated F1 Score of each model is inconsistent with the value in the table of the manuscript.

5) In lines 303-307 of the manuscript, the model implemented on BERT and ALBERT are discussed, and the conclusion is that ALBERT is not as effective as BERT. But,ALBERT addresses the issues of the large model size and high parameter count in BERT, aiming to reduce computational resource consumption and improve training efficiency. It is suggested to increase discussion on training resources and improving training efficiency.

6) It is suggested to add ablation experiments to demonstrate the contribution of each part of the model.

7) Figure 6 lists the impact of each parameter on the results, and Tables 6, 7, 8, and 9 also list the result values. There are overlaps in these two parts. Moreover, it is suggested to include a separate section in the experimental results and analysis to discuss the impact of parameters on the results, so that the structure is clearer.

Validity of the findings

The conclusion of the manuscript is clear and highlights the novelty of the proposed model. To further improve the submission, I suggest:
1)The existing experiments are insufficient to support the conclusions in lines 353-355 of the manuscript.
2)To add more details about the limitations of the proposed model in this manuscript, as well as future research directions.

Additional comments

none

---

## Round 0.2 · accepted · Accept

The original reviewers were invited but did not respond. I checked the revised version and believe that the authors have addressed the issues proposed by the reviewers. Thus, I recommend accepting this article. Thanks for the efforts of the authors.